# Epimeric Mixture Analysis and Absolute Configuration Determination Using an Integrated Spectroscopic and Computational Approach—A Case Study of Two Epimers of 6-Hydroxyhippeastidine

**DOI:** 10.3390/molecules28010214

**Published:** 2022-12-26

**Authors:** Ngoc-Thao-Hien Le, Tom Vermeyen, Roy Aerts, Wouter A. Herrebout, Luc Pieters, Emmy Tuenter

**Affiliations:** 1Natural Products & Food Research and Analysis (NatuRA), Department of Pharmaceutical Sciences, University of Antwerp, Universiteitsplein 1, 2610 Antwerp, Belgium; 2Molecular Spectroscopy (MolSpec), Department of Chemistry, University of Antwerp, Groenenborgerlaan 171, 2020 Antwerp, Belgium

**Keywords:** 6-hydroxyhippeastidine, epimeric mixture analysis, DFT calculation, OR/ORD, ECD, VCD, CASE

## Abstract

Structural elucidation has always been challenging, and misassignment remains a stringent issue in the field of natural products. The growing interest in discovering unknown, complex natural structures accompanies the increasing awareness concerning misassignments in the community. The combination of various spectroscopic methods with molecular modeling has gained popularity in recent years. In this work, we demonstrated, for the first time, its power to fully elucidate the 2-dimensional and 3-dimensional structures of two epimers in an epimeric mixture of 6-hydroxyhippeastidine. DFT calculation of chemical shifts was first performed to assist the assignment of planar structures. Furthermore, relative and absolute configurations were established by three different ways of computer-assisted structure elucidation (CASE) coupled with ORD/ECD/VCD spectroscopies. In addition, the significant added value of OR/ORD computations to relative and absolute configuration determination was also revealed. Remarkably, the differentiation of two enantiomeric scaffolds (crinine and haemanthamine) was accomplished via OR/ORD calculations with cross-validation by ECD and VCD.

## 1. Introduction

Natural products chemistry is one of the oldest research areas. After more than a century of development, a wide variety of hit compounds are still being isolated from natural sources [1]. Thanks to the continuous advancement of chromatographic and spectroscopic techniques, scientists are able to explore minor and/or complex components. In recent years, the application of quantum chemical calculations has facilitated accurate, high-speed, theoretical methods to predict 2D and 3D molecular structures [2]. Reliable methods were established and optimized to compute NMR chemical shifts and coupling constants with minimal computational cost [2,3]. Regarding 3D structural identification, a breakthrough was first made by the Goodman group, introducing the CP3 parameters in 2009, followed by the DP4 probabilistic method in 2010 [4,5]. While the application of the former method is limited, the latter was successfully applied in many cases to confirm or correct the relative configuration of natural products of varying complexity [6,7,8]. More recently, the Sarotti group introduced two modified probabilistic methods based on the mathematical core of the DP4 probability: DP4+ and *J*-DP4 [9,10]. Many successful applications of the DP4+ probability method have already been reported in the field of natural products [7,11]. Most recently, the DP5 method was published by the Goodman group in 2022 [12]. Nonetheless, the computational effort associated with these methods can be significant, especially in the case of large and flexible molecules. In order to reduce this large workload, several (semi-)automatic procedures were developed, using freely accessible programming languages (Python, Bash and R), Excel (DP4+ and *J*-DP4 templates), machine learning and deep learning [9,13,14]. Software applications implementing Computer-Assisted 3D Structure Elucidation (CASE-3D) were also developed recently, such as the StereoFitter from Mestrelab MNova, CMC-se from Bruker, Structure Elucidator from ACD labs and Logic Structure Determination (LSD) from Nuzillard and Plainchont [15,16,17,18]. The considerable efforts made by theoretical chemists to introduce these methods to other disciplines, such as organic synthesis, biosynthesis and natural products chemistry, have resulted in a rapidly increasing number of citations within the last five years [2,11,19,20].

Over the last two decades, quantum chemical predictions of optical rotation (OR) or optical rotation dispersion (ORD) have become a fast and reasonably reliable tool to determine absolute configurations [21,22,23]. Several successful applications of this method were published recently, as such or in combination with other spectroscopic techniques such as electronic circular dichroism (ECD), vibrational circular dichroism (VCD) and/or Raman optical activity (ROA) [24,25,26]. As a result of these theoretical calculations, the information obtained from OR/ORD was extended and is no longer limited to just providing a single value and defining a compound as dextrorotatory or levorotatory. By comparing the computed and experimental OR/ORD values, possibly at multiple wavelengths, the sign of the rotation can be translated into an enantiomer described using R/S nomenclature.

In the structural elucidation of natural products field, misassignment remains a stringent and commonly unnoticed issue [27,28,29,30,31,32]. In the present study, a rational and general strategy using a combination of empirical and computational data was applied, combining all the above-mentioned advancements (Figure 1). A major asset of this workflow lies in combining these various methods. Thus, each structural property of the molecule can be extracted from multiple sources, making the elucidation more robust and minimizing the probability of misassignment. For the first time, the effectiveness of the workflow was tested on an epimeric mixture of two 6-hydroxyhippeastidines purified from the plant species *Hymenocallis littoralis* (Amaryllidaceae). The crinine-type and haemanthamine-type scaffolds, widely distributed in the Amaryllidaceae family, are characterized by the arylhydroindole ring system and are composed of over 50 members in nature [33,34]. In this work, two epimers of 6-hydroxyhippeastidine, belonging to the crinine-type subgroup, were studied. Their 2D chemical structures were elucidated based on the basis of extensive 1D and 2D NMR spectroscopy and HRMS data. Relative and absolute configurations were determined by pooling the results obtained with optical rotation, nuclear Overhauser effect (NOE), DP4+, ECD and VCD analyses.

## 2. Results

### 2.1. 2-Dimensional Structure Elucidation

Compounds **1** and **2** were isolated as an inseparable epimeric mixture (ratio ~3.15:1, averaged from the ratios of two corresponding pairs of the H-7 and H-6 ^1^H-NMR signals of the two compounds, as shown in Figure 2 and Figure 3). The crinine- or haemanthamine-type skeleton was first proposed for compounds **1** and **2** based on a comparison with the NMR data of haemanthidine in the literature [33]. Haemanthidine is a well-known mixture of two epimers and is widely distributed in the Amaryllidaceae family. In the case of haemanthidine, both 6-epimers were present (adopting the same atom numbering system commonly used in the literature), as deduced from the difference in the chemical shift of H-6: the H-6 of the major epimer was found at 5.02 ppm, and the H-6 of the minor epimer was found at 5.65 ppm in (CD_3_)_2_SO [34]. A large similarity was observed for the epimeric center of compounds **1** and **2**: the H-6 of the major epimer resonated at 4.72 ppm, while that of the minor epimer resonated at 5.43 ppm in (CD_3_)_2_SO. Continuing to use haemanthidine as the reference point, two noticeable differences were observed by comparing the ^1^H NMR spectra: Firstly, a signal around 5.95 ppm with an integration of the two was present in the case of haemanthidine but was absent in the case of compounds **1** and **2**, indicating the absence of the methylenedioxy moiety. Secondly, in the downfield region, four signals corresponding to aromatic and olefinic hydrogens were observed in the ^1^H NMR spectrum of haemanthidine, whilst only one aromatic hydrogen was found for compounds **1** and **2**. This infers that the double bond present in haemanthidine was absent in compounds **1** and **2** and that compounds **1** and **2** possess one more substituent on the benzene ring compared to haemanthidine. This assumption was supported by the appearance of two additional methoxy signals around 3.65–3.75 ppm and five additional aliphatic hydrogens in the upfield region from 1.00 to 2.50 ppm in the ^1^H NMR spectrum of the mixture of epimers. Next, the 2D spectra were inspected. Five methylene and three methoxy groups were first determined in the ^1^H NMR and HQSC spectra. Furthermore, two spin systems were observed in COSY: the first one for H-1 (2H), H-2 (2H), H-3 (1H) and H-4 (2H) and the second one for H-11 (2H) and H-12 (2H) (see Figure 4). A strong HMBC signal between the methoxy group (H at 3.24 ppm) and C-3 (77.6 ppm) inferred that they were connected. Notably, since the NMR signals of compound **2** were much less intense than those of compound **1** in the mixture, an additional long-range HMBC spectrum was recorded to observe the 4- and 5-bond HMBC correlations. This led to the observation of a peak between the H-6 and C-10b of compound **2** and to the identification of the signal corresponding to C-10b.

With regard to the aromatic moiety, the proton signal at 6.32 ppm only showed an HMBC correlation with C-6 and not with any other CH or CH_2_ group of the aliphatic moiety, indicating that this proton signal was present at position 7. After assigning 3-OMe, two methoxy-groups remained, which were linked to the benzene ring. The position of the 8-OMe was confirmed by a NOESY cross-peak between H-7 and 8-OMe (Figure 5). Nonetheless, the HMBC spectrum confirmed that the last methoxy group was present at position 10, given the correlations of H-6 with C-6a, C-7, C-8, C10a and one additional signal of the methoxylated carbon, which most probably would be present in position 10. Hence, the hydroxy group would be present in the only remaining aromatic position, position 9, and the NMR signal of C-9 was found at 147.7 ppm (see Figure 6, structure a). Combining all this information and using HMBC and NOESY correlations to procure the assignments, compounds **1** and **2** were initially identified as the two 6-epimers of 6-hydroxyhippeastidine (see Figure 6, structure a). 

However, surprisingly, the calculated chemical shifts of the proposed structures (Figure 6, structure a) were not in full agreement with the experimental data, in particular regarding the aromatic moiety. As listed in Table 1 and Table 2, alarmingly high discrepancies (~10 ppm) were found for C-9 and C-10a, while their surrounding carbons showed 4–6 ppm errors (C-6a, C-8 and C-9). This raises a question about the assignment at the aromatic part of compounds **1** and **2** [11]. Based on the differences in shielding intensities of C-9 and C-10 (C-9 is more deshielded than C-10, according to the experimental data, while for the computations the opposite was observed), a new structural proposition (structure b) was considered, in which the C-9 and C-10 substituents were switched (Table 1 and Table 2 and Figure 6). Finally, chemical shifts of structure b were computed following the approach reported by the authors [31]. Indeed, computed chemical shifts of structure b showed an utter resemblance to the experimental values, with a CMAE of 1.04 ppm and a maximum outlier of 2.83 ppm for the carbons of compound **1** (Table 1) and a CMAE of 1.09 ppm and a maximum outlier of 2.94 ppm for the carbons of compound **2** (Table 2). Hence, structure b was confirmed to be correct, and compounds **1** and **2** were the two 6-epimers of structure b, which were reported in the plant species *Zephyranthes candida* by Shitara et al. (6α- and 6β-hydroxyhippeastidine) [35]. Remarkably, the HMBC correlation between H-6 and C-9 is a *J*_5_ interaction, which is unexpected and rarely observed, while the *J*_4_ interaction between H-6 and C-10 was absent in both the regular and long-range HMBC spectra (Figure 6, structure b).

### 2.2. Relative Configuration Determination

Further analysis of the NOE correlations (Figure 5) yielded the proposed relative configuration of the two compounds: (3***R***,4a***R***,6***R***)-6α-hydroxyhippeastidine or its enantiomer for compound **1** (major epimer) and (3***R***,4a***R***,6*S*)-6β-hydroxyhippeastidine or its enantiomer for compound **2** (minor epimer). Briefly, starting from the epimeric center at C-6, the H-6 (4.72 ppm) of compound **1** correlated with the H-12 of the ethanobridge, while the H-6 (5.43 ppm) of compound **2** did not, indicating that the H-6 of compound **2** is located on the same side of the ethanobridge (β-orientation). On the other hand, the H-6 (5.43 ppm) of compound **2** correlated with the H-4a (2.95 ppm), while the H-6 (4.72 ppm) of compound **1** did not, indicating that the H-4a of compounds **1** and **2** is on the opposite side of the ethanobridge and that the H-6 (5.43 ppm) of compound **2** and the H-4a were located on the same side (α-orientation). Finally, the H-4a correlated with the other hydrogens on the α-face (H-1α, H-2α, H-4α and 3-OMe), but not with the H-3, indicating the β-configuration of the H-3.

Before conducting DP4+ as the next step, it is noteworthy that the crinine type and haemanthamine type are two enantiomeric skeletons, defined by the orientation of the 5,10b-ethanobridge moiety: in the case of the haemanthamine skeleton, the ethanobridge is α-oriented, and in the case of crinine-type alkaloids, the ethanobridge is β-oriented [33,35]. However, computed chemical shifts of enantiomers will be identical, and, thus, only calculation of one set of enantiomers is required. The set of crinine-type diastereomers was selected to perform the calculations in this DP4+ probability assessment. Given the presence of the ethanobridge, which in the crinine-type skeleton is assigned the β-configuration, the number of stereogenic centers to consider is reduced from five (including the N at position 5) to three. Thus, eight diastereomers were computed for compounds **1** and **2**. According to Table 3, the DP4+ probabilities were in complete accordance with the preceding NOESY correlations. Indeed, the computed chemical shifts of the two proposed diastereomers showed the highest resemblance to the experimental chemical shifts. Thus, the relative configurations of compounds **1** and **2** were defined by two means: NOE correlations and DP4+ probability. The relative configuration assignment is clearly robust, leaving only the absolute configurations to be assigned.

### 2.3. Absolute Configuration Determination

Using the relative configurations established in the previous section, the optical rotations were computed for both compounds. As indicated in Table 4, the optical rotation values were computed for four configurations, representing two pairs of enantiomers. More specifically, the diastereomers with the RRRRR and RRRSR configurations belong to the crinine-type skeleton, whilst the compounds with the SSSSS and SSSRS configurations belong to the haemanthamine-type skeleton. Since compounds **1** and **2** were obtained as a mixture with a ratio of 3.15:1.00, the calculated ORs had to be adjusted accordingly. 

In the case of the ORs computed by the Pople (6-311++G(3df,2dp)) basis set:-the averaged OR of the crinine-type = $\frac{80.3*3.15\;+\left(-54.2\right)*1}{\left(3.15+1\right)}$ = 47.9;-the averaged OR of the haemanthamine-type = $\frac{\left(-80.3\right)*3.15\;+54.1*1}{\left(3.15+1\right)}$ = −47.9.


In the case of the ORs computed by the Dunning (aug-cc-pVTZ) basis set:-the averaged OR of the crinine-type = $\frac{80.1*3.15\;+\left(-53.7\right)*1}{\left(3.15+1\right)}$ = 47.8;-the averaged OR of the haemanthamine-type = $\frac{\left(-80.1\right)*3.15\;+53.5*1}{\left(3.15+1\right)}$ = −47.9.

After comparison of the above computed ORs with the experimental value (43.7), it is apparent that compounds **1** and **2** are of the crinine type. As a result, the absolute configuration of compound **1** is (3***R***,4***R***,5***R***,6***R***,10b***R***) and that of compound **2** is (3***R***,4***R***,5***R***,6*S*,10b***R***). More interestingly, this result led to the conclusion that the OR/ORD calculation is capable of differentiating the crinine-type skeleton from the haemanthamine-type skeleton. As a technical remark, although the results obtained with the 6-311++G(3df,2dp) and aug-cc-pVTZ basis sets show a slight difference, the two basis sets both provided decent and conclusive results.

The assignment of the absolute configuration of compounds **1** and **2** was confirmed by electronic circular dichroism (ECD) and vibrational circular dichroism (VCD) analyses. To this end, an ECD spectrum was recorded, following the conventional way to distinguish crinine-type and haemanthamine-type skeletons. In 1996, Wagner et al. reported a study on ECD for one crinine-type (crinidine) and two haemanthamine-type alkaloids (vittatine and 11-hydroxyvittatine), which later on has been used as an empirical rule for the determination of crinine- and haemanthamine-type skeletons [36]. Crinidine shows a maximum Cotton effect around 240 nm and a minimum Cotton effect around 290 nm. Vittatine and 11-hydroxyvittatine, on the other hand, display opposite Cotton effect patterns. Figure 7 demonstrates the ECD spectrum recorded for the mixture of compounds **1** and **2**, which shows similar Cotton effects to crinidine. In comparison with other known crinine-type alkaloids, a larger similarity in Cotton patterns was observed for aulicine, an Amaryllidaceae alkaloid that is structurally close to compounds **1** and **2** [37]. Hence, it was confirmed that compounds **1** and **2** possess the crinine-type skeleton and that the results obtained from the OR/ORD calculation were the same as those from ECD.

The results of the VCD analyses are summarized in Figure 8. In the first step of the calculations, Boltzmann weighted IR and VCD calculations were performed at the B3LYP/6-31++G(d,p) level for both the major and minor epimers. This approach is similar to the one used recently to determine the absolute stereochemistry of the antibiotic polyketide tatiomicin [38]. Subsequently, the resulting spectra were combined, and a weighted IR and VCD spectrum was obtained using the 3.15:1.00 ratio, as determined from the NMR intensities. The experimental data obtained, the calculated IR and VCD spectra for the epimers involved and the resulting spectra for the mixture are shown in Figure 8. A good agreement supporting the absolute configurations of compounds **1** and **2** proposed above is found between the experimental data in the upper panels and the final calculated data shown in the lower panels.

## 3. Discussion

The general workflow demonstrated in Figure 1 was successfully applied to the epimeric mixture in this study. Thus, this work showed the added value of quantum chemical calculations to predict chemical shifts in the structure elucidation of the 2D structure, in addition to the interpretation of the experimental 1D and 2D NMR data. When the discrepancies between the experimental and calculated data are >1 ppm for the proton data and >10 ppm for the carbon data, caution should be taken [2,11]. The experimental OR value of the epimeric mixture was successfully reproduced by quantum chemical calculations when taking into account the ratio of the two epimers in the mixture.

Whilst ECD still prevails and vibrational optical activity (VOA)-based methods (VCD and ROA) are gaining popularity, in our opinion OR/ORD is underrated and is favorable in terms of simplicity, accessibility and cost-effectiveness for the absolute configuration determination of relatively rigid, small chiral molecules in natural products research. OR/ORD can be applied to support ECD, VCD and ROA results, while in some cases OR/ORD data alone are even sufficient to draw conclusions regarding the absolute configuration of a certain compound [39,40,41,42]. Moreover, interpretation of OR/ORD data is straightforward, which makes this method suitable for application as a routine practice in natural products chemistry. Nevertheless, it is vital to emphasize that there is no one-size-fits-all technique, and the required type(s) of spectroscopic methods should be assessed on a case-by-case basis [41]. From our perspective, the following five criteria are essential for the proper application of OR/ORD for absolute configuration determination: (1)—experimental data are obtained for the purified compound (or pure mixture) in an appropriate amount depending on instrumental sensitivity and molecular properties; (2)—NOE correlations are used for relative configuration determination or as a filter to reduce the pool of computed diastereomers [43]; (3)—identical or strictly equivalent conformer pools of a pair of enantiomers are required; (4)—at least two different types of basis sets of a sufficiently large size should be used for computing OR/ORD; (5)—the need for other chiroptical/spectroscopic techniques should be considered when experimental OR/ORD values are close to zero, as is the case for racemic mixtures.

## 4. Materials and Methods

### 4.1. General Experimental Procedure

NMR spectra were recorded on two different NMR systems (400 MHz): an Avance Nanobay III and a DRX-400 system (Bruker BioSpin, Rheinstetten, Germany). NMR data processing was performed using TopSpin v.4.0.6 from Bruker. Accurate mass measurements were conducted using a Xevo G2-XS QToF mass spectrometer (Waters, Milford, MA, USA) coupled to an Acquity LC system (an Acquity HSS T3 UPLC column (100 × 2.1 mm, 1.8 μm) was used). For NMR experiments, chloroform-d (CDCl_3_—99.8% D) and dimethyl sulfoxide-d_6_ ((CD_3_)_2_SO—99.9% D) was purchased from Sigma-Aldrich (Merck, Germany).

Optical rotation was measured on a Jasco P-2000 spectropolarimeter (Easton, MD, USA), equipped with Spectra Manager^TM^ software, at the sodium D-line wavelength (589.3 nm). 

ECD spectra were recorded using a Chirascan^TM^ -Plus spectrophotometer (Applied Photophysics Ltd., Leatherhead, UK), equipped with a SUPRASIL^R^ quartz cell (Hellma BeNeLux, Kruibeke, Belgium) with a path length of 2 mm. Measurement conditions and settings were as follows: continuous flushing of the instrument with nitrogen gas (4 L.min^−1^ flow rate), temperature: 20 °C, bandwidth: 1 nm and acquisition time: 1 s.nm^−1^. 

The experimental IR and VCD spectra were recorded on a Bruker Invenio FTIR spectrometer equipped with a PMA50 accessory (Bruker Optics, Ettlingen, Germany). All measurements were performed in CDCl_3_ at a concentration of 0.12 M. A cell with 100 μm path length and BaF_2_ windows was used. Both the sample and the solvent spectrum were recorded with a resolution of 4 cm^−1^, totaling 30,000 scans each, with the PEMs optimized at 1400 cm^−1^. The final baseline-corrected VCD spectrum was obtained through subtraction of the solvent spectrum.

Analytical grade solvents of methanol (MeOH), acetonitrile (ACN), dichloromethane and formic acid were purchased from Sigma-Aldrich (St. Louis, MO, USA) and Acros Organics (Geel, Belgium).

### 4.2. Isolation and Purification

The mixture of two epimers was isolated from the bulbs of *Hymenocallis littoralis*. Briefly, dried plant material was macerated with 80% MeOH to obtain crude extract. The crude extract was acidified by HCl 5% to pH < 3 and, thereafter, partitioned with dichloromethane (x3). The crude dichloromethane extract was fractionated by flash chromatography to obtain 15 fractions. The two epimers were purified from fraction 10 by preparative HPLC using the following conditions: column Kinetex C18 (250 × 4.6 mm, 5 µm), solvents (0.1% FA in H_2_O and ACN), flow rate: 3 mL/min and fraction trigger: 336.3 *m/z*. 

### 4.3. Computational Details

Conformational search was performed with PCMODEL (version 10.0) using the Monte Carlo algorithm and Merck Molecular Force Field (MMFF94). Two energy windows (8 and 7 kcal.mol^−1^) were set for two consecutive conformational search cycles. Next, the Density Functional Theory (DFT) method, utilizing the exchange-correlation functional B3LYP (hybrid three-parameter Becke–Lee–Yang–Parr), was applied for the following calculations. All resulting conformers were subjected to geometry optimization using B3LYP/6–31G (d) level of theory in gas-phase, and isotropic shielding tensors were computed at the PCM//mPW1PW91/6-311+G(d,p) level of theory by the GIAO (gauge-independent atomic orbital) method. Boltzmann-averaged shielding tensors were used as input for carrying out the DP4+ probability. The DP4+ probability method using Bayes’ theorem was applied in order to provide additional proof of relative configurations.

After dereplication, Boltzmann averaging was applied using the sum of electronic and thermal free energies at 298.15 K, and only conformers with energies within an energy window of 2.5 kcal.mol^−1^ from the global minimum were considered as contributing to the population. Geometry optimization, frequency, shielding tensor and specific rotation calculations were performed by Gaussian16. Avogadro (version 1.2.0) was used for the visualization of computed outputs. Detailed information about results of DP4+ analysis, Boltzmann distributions and 3D coordinates of contributing conformers can be found in Appendix A.

Specific optical rotations were computed for the sodium D-line wavelength (589.3 nm). All conformers were optimized at the B3LYP/6-31++G(d,p) level of theory. The polarizable continuum model (PCM) was selected to improve the OR calculation accuracy (chloroform was used for the solvation model). The DFT method using the B3LYP functional combined with the two following basis sets was applied: a Pople 6-311++G (3df,2dp) and a Dunning (aug-cc-pVTZ), as recommended by Yu et al. (2012) and Stephens et al. (2001), respectively [44,45].

## 5. Conclusions

Epimeric mixture analysis has always been challenging in natural products research, since the identification and isolation of single compounds is highly demanding. This work served as a case study to explore the power of combining various spectroscopic methods and computational chemistry in the structural elucidation procedure, from planar structure to absolute configuration, of an epimeric mixture. In addition, the potential of OR/ORD simulation is highlighted for various applications. 

Another interesting finding from our experiments is that crinine and haemanthamine skeletons (two enantiomeric scaffolds) can be differentiated by OR/ORD calculation, without the need for an ECD and/or VCD experiment. This will aid in simplifying the structural elucidation process of compounds possessing crinine or haemanthamine skeletons. Hence, this could be considered as a new benchmark for the distinguishment of the two skeleton types, since the publication of Wagner et al. in 1996 [36]. For the future, since this work only dealt with an example, it would be interesting to investigate a library of crinine-type and haemanthamine-type analogues to further confirm this finding.

## Figures and Tables

**Figure 1 molecules-28-00214-f001:**
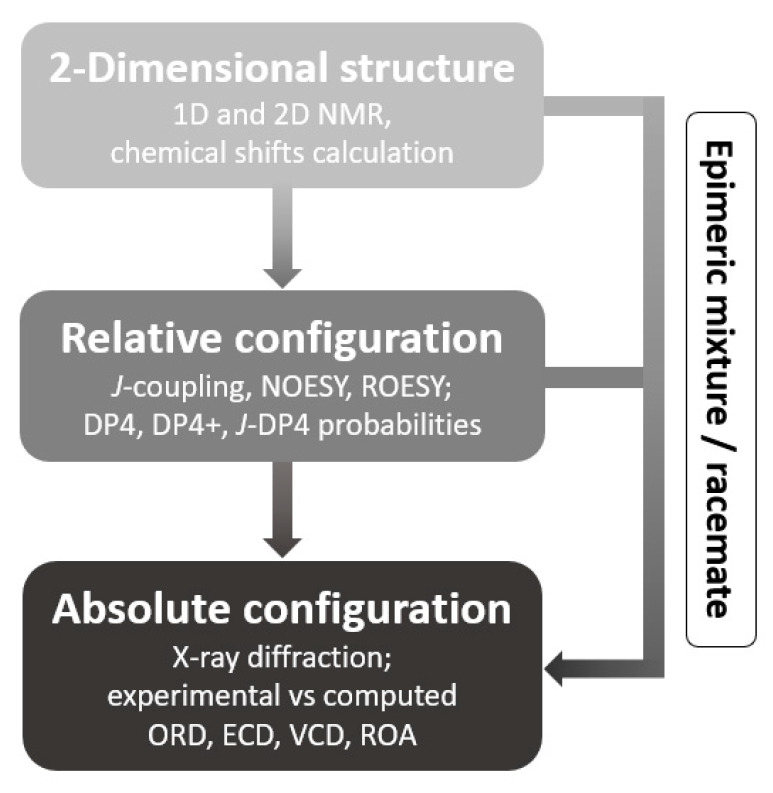
General Structural Elucidation Workflow.

**Figure 2 molecules-28-00214-f002:**
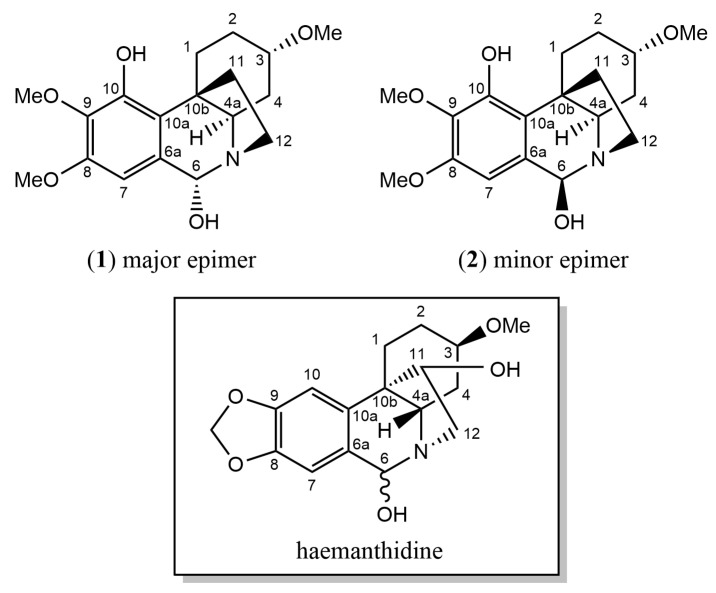
Structures of 6α-hippeastidine (compound **1**), 6β-hippeastidine (compound **2**) and haemanthidine.

**Figure 3 molecules-28-00214-f003:**
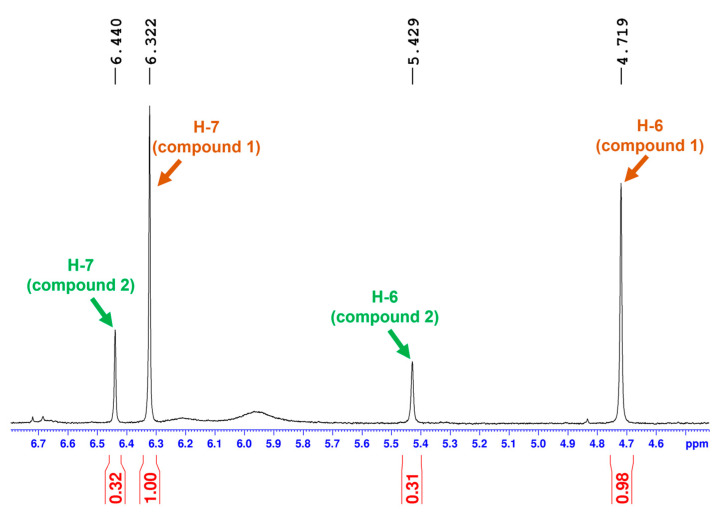
Epimeric ratio of the mixture of compounds **1** and **2**, deduced from the ^1^H-NMR spectrum.

**Figure 4 molecules-28-00214-f004:**
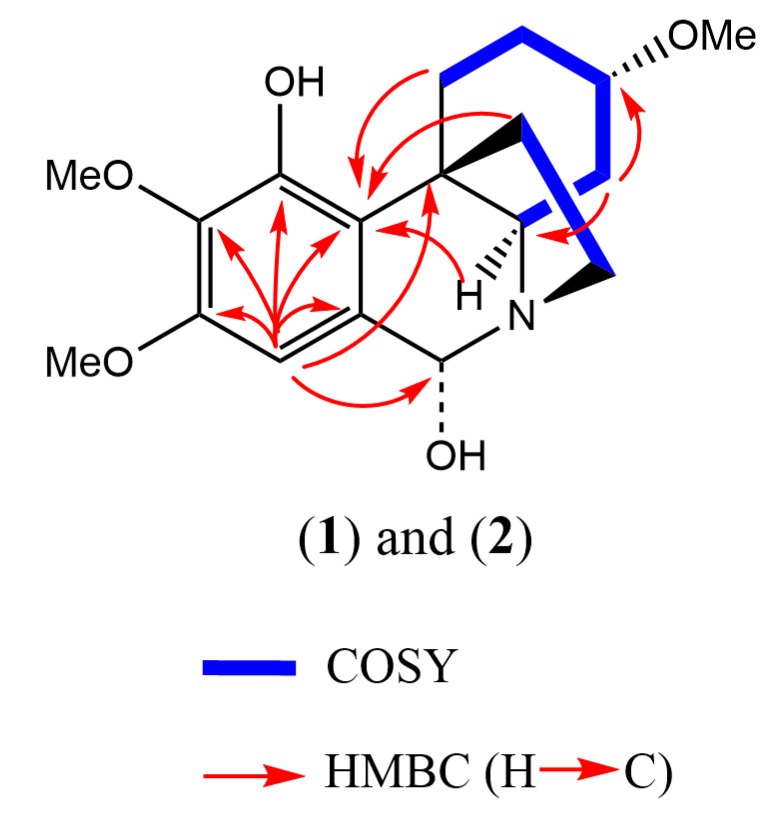
Key HMBC and COSY correlations for two 6-epimers of 6-hydroxyhippeastidine.

**Figure 5 molecules-28-00214-f005:**
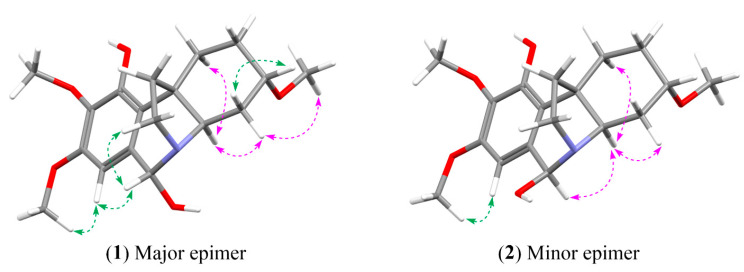
Selected NOESY correlations for two 6-epimers of 6-hydroxyhippeastidine. Green arrows: NOESY correlations in front of the plane (β-orientation); pink arrows: NOESY correlations behind the plane (α-orientation).

**Figure 6 molecules-28-00214-f006:**
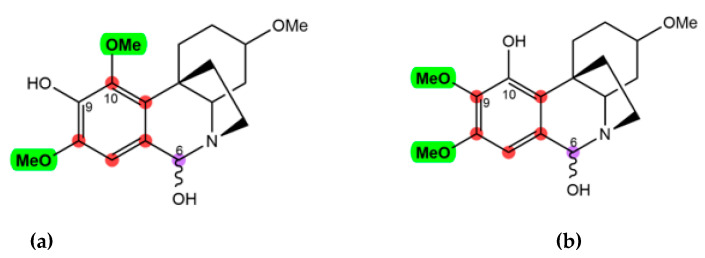
Observed HMBC correlations between H-6 (in purple) and carbons of the aromatic ring (in red) of compounds **1** and **2**: (**a**) incorrect assignment and (**b**) correct assignment.

**Figure 7 molecules-28-00214-f007:**
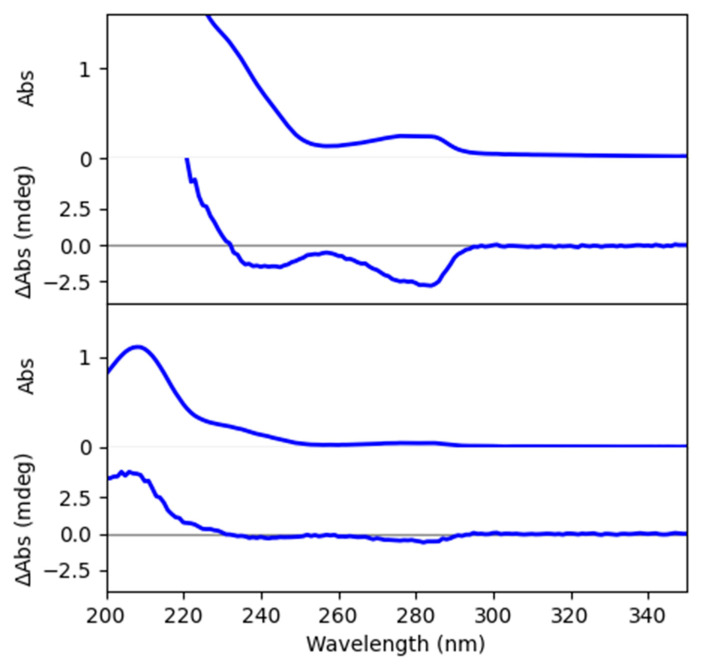
UV and ECD spectra of the mixture of compounds **1** and **2**. Top: spectra recorded for a high concentrated sample. Bottom: spectra recorded for a low concentrated sample.

**Figure 8 molecules-28-00214-f008:**
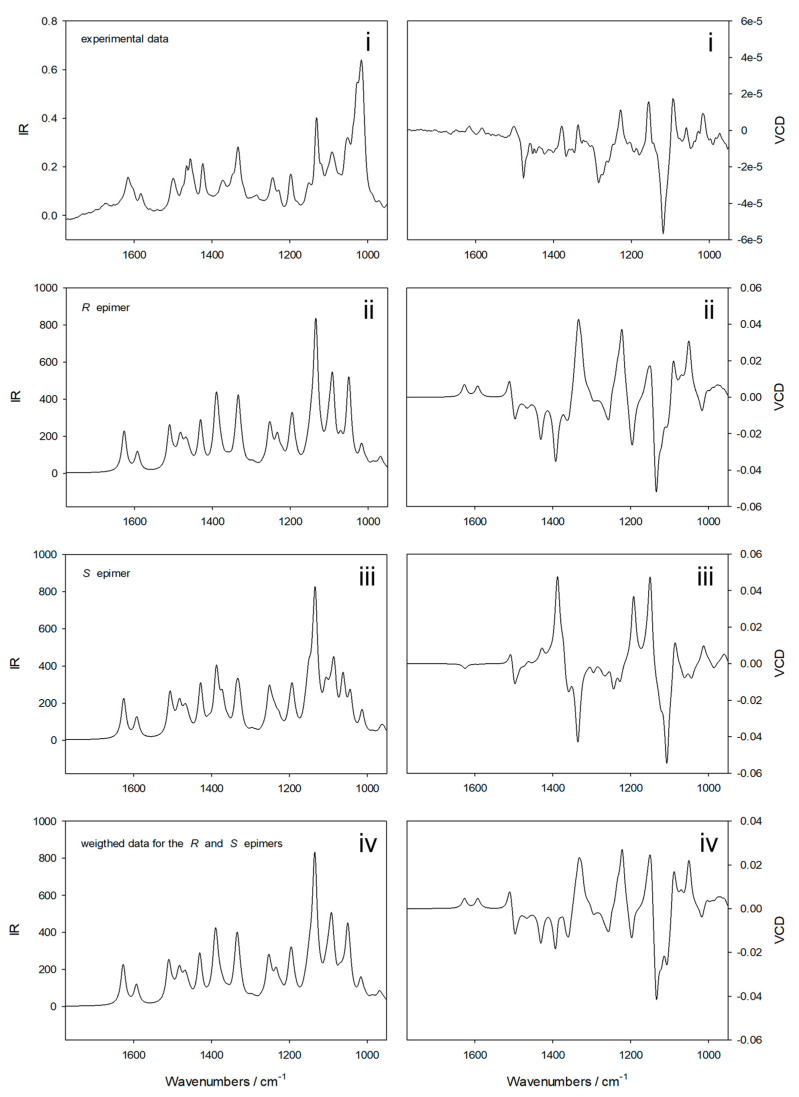
Experimental IR and VCD spectra confirming the absolute configurations of compounds **1** and **2**. From top to bottom, the data shown refer to (**i**) the experimental data obtained for the sample, (**ii**) the Boltzmann weighted IR and VCD spectra for the (3*R*,4*R*,5***R***,6*R*,10b***R***) epimer, (**iii**) the Boltzmann weighted IR and VCD spectra for the (3*R*,4*R*,5***R***,6*S*,10b***R***) epimer, and (**iv**) the weighted IR and VCD spectra obtained using the 3.15:1.00 ratio for major and minor epimers as derived from the NMR data.

**Table 1 molecules-28-00214-t001:** Experimental and computed NMR data (400 MHz, (CD_3_)_2_SO) of compound **1** (major epimer).

	Experimental		Calculated			
			Structure Proposal 1a		Structure Proposal 1b	
Position	δ_C_, Type	δ_H_ (*J* in Hz)	δ_C_	δ_H_	δ_C_	δ_H_
1α	27.1, CH_2_	1.55	27.7	1.60	27.9	1.63
1β	27.1, CH_2_	3.12 *	27.7	2.99	27.9	2.94
2α	27.9, CH_2_	1.90 m	27.9	1.90	28.2	1.88
2β	27.9, CH_2_	1.30 m	27.9	1.32	28.2	1.29
3	77.6, CH	3.10 *	76.7	3.01	77.1	3.03
4α	33.7, CH_2_	1.82 m	33.2	1.76	33.5	1.76
4β	33.7, CH_2_	1.05 q (11.8)	33.2	1.10	33.5	1.08
4a	61.3, CH	3.07 *	62.4	2.99	62.5	3.01
6	88.8, CH	4.72 s	88.9	4.68	89.0	4.75
6a	132.0, C	-	127.4	-	132.8	-
7	103.8, CH	6.32 s	106.2	6.45	103.5	6.36
8	150.9, C	-	146.3	-	151.4	-
9	135.8, C	-	144.6	-	134.8	-
10	147.7, C	-	140.9	-	147.9	-
10a	127.3, C	-	137.2	-	128.4	-
10b	43.3, C	-	46.2	-	46.1	-
11	33.3, CH_2_	2.06 m	34.8	2.11	34.5	2.08
11	33.3, CH_2_	1.51 *	34.8	1.61	34.5	1.58
12	47.3, CH_2_	3.12 *	47.5	3.11	47.8	3.13
12	47.3, CH_2_	2.52	47.5	2.55	47.8	2.56
3-OMe	55.3, CH_3_	3.24 s	53.2	3.20	53.5	3.22
8-OMe	55.9, CH_3_	3.72 s	52.9	3.68	53.2	3.65
9-OMe	60.6, CH_3_	3.63 s	55.7	3.70	57.8	3.74
CMAE			3.0	0.06	1.04	0.06
Max. outlier			9.9	0.13	2.83	0.18

* overlapping signals.

**Table 2 molecules-28-00214-t002:** Experimental and computed NMR data (400 MHz, (CD_3_)_2_SO) of compound **2** (minor epimer).

	Experimental		Calculated			
			Structure Proposal 2a		Structure Proposal 2b	
Position	δ_C_, Type	δ_H_ (*J* in Hz)	δ_C_	δ_H_	δ_C_	δ_H_
1α	27.1	1.55	27.6	1.63	28.0	1.66
1β	27.1	3.12 *	27.6	2.98	28.0	2.95
2α	27.9	1.90 m	27.8	1.89	28.0	1.88
2β	27.9	1.30 m	27.8	1.32	28.0	1.29
3	77.3	3.10 *	76.6	3.01	76.8	3.03
4α	33.9	1.88 m	33.4	1.85	33.7	1.84
4β	33.9	1.13 q (11.8)	33.4	1.16	33.7	1.16
4a	66.3	2.95	67.0	2.85	67.1	2.88
6	86.8	5.43 s	87.8	5.37	87.9	5.42
6a	133.1	-	128.5	-	134.0	-
7	102.3	6.44 s	104.3	6.63	101.7	6.50
8	150.8	-	146.3	-	151.3	-
9	135.6	-	144.4	-	134.6	-
10	147.4	-	140.7	-	147.7	-
10a	126.5	-	136.0	-	127.0	-
10b	44.3	-	47.2	-	47.2	-
11	35.0	2.13 m	37.1	2.20	36.6	2.17
11	35.0	1.48 *	37.1	1.54	36.6	1.52
12	41.9	3.17 *	41.6	3.20	42.2	3.21
12	41.9	2.76	41.6	2.69	42.2	2.72
3-OMe	55.3	3.24 s	53.1	3.21	53.4	3.23
8-OMe	55.9	3.72 s	52.8	3.69	53.1	3.65
9-OMe	60.6	3.63 s	55.6	3.70	57.7	3.74
CMAE			3.1	0.06	1.09	0.06
Max. outlier			9.5	0.16	2.94	0.17

* overlapping signals.

**Table 3 molecules-28-00214-t003:** DP4+ probabilities of two 6-epimers of 6-hydroxyhippeastidine. Two bold letters R stand for the fixed configuration of the 5,10b-ethanobridge with respect to the crinine-type skeleton (3,4,5***R***,6,10b***R***).

Compound 1		Compound 2	
Diastereomer	Probability (%)	Diastereomer	Probability (%)
RR**R**R**R**	100	RR**R**R**R**	0
RR**R**S**R**	0	RR**R**S**R**	100
RS**R**R**R**	0	RS**R**R**R**	0
RS**R**S**R**	0	RS**R**S**R**	0
SR**R**R**R**	0	SR**R**R**R**	0
SR**R**S**R**	0	SR**R**S**R**	0
SS**R**R**R**	0	SS**R**R**R**	0
SS**R**S**R**	0	SS**R**S**R**	0

**Table 4 molecules-28-00214-t004:** Experimental and computed optical rotation values for two 6-epimers of 6-hydroxyhippeastidine (compounds **1** and **2**).

	Diastereomer	OR	Theory Level
Experimental		43.7	
Computed	RRRRR	80.3	B3LYP/6-31++G(d,p)// 6-311++G(3df,2dp)
	RRRSR	−54.2	
	SSSSS	−80.3	
	SSSRS	54.1	
	RRRRR	80.1	B3LYP/6-31++G(d,p)// aug-cc-pVTZ
	RRRSR	−53.7	
	SSSSS	−80.1	
	SSSRS	53.5	

## Data Availability

Data are contained within the article or the Appendix A.

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
