# Peer review of "Epimeric Mixture Analysis and Absolute Configuration Determination Using an Integrated Spectroscopic and Computational Approach—A Case Study of Two Epimers of 6-Hydroxyhippeastidine"

_molecules, 2022, doi:10.3390/molecules28010214_

Round 1

Reviewer 1 Report

Please see the attached pdf document for complete review report and comments.

Reviewer 2 Report

The studies developed in this work contribute for the study of structural elucidation in the field of natural products. The study is interesting and suitable for publication, however, minor revision is required before the paper can be considered for publication on Molecules.

 Below are the comments of the reviewer:

 1) I suggest improving the abstract. Add the results found;

 2) Substances studied at work need to be better presented in the introduction;

 3) In the results and discussion I suggest further discussing the differences between the experimental and theoretical results found.

Round 2

Reviewer 1 Report

The paper can be accepted in its current form.